# Recurrent Melanoma in a Patient with Chronic Lymphocytic Leukemia (CLL) Presenting with an Apparent Co-Existing NRAS and BRAF Mutation: A Diagnostic and Treatment Conundrum

**DOI:** 10.3390/ijms26031029

**Published:** 2025-01-25

**Authors:** Giuliana G. Berardi, Jabbar Muthanna, Y. Lynn Wang, Anthony J. Olszanski

**Affiliations:** 1Department of Medical Oncology, Fox Chase Cancer Center, Philadelphia, PA 19111, USA; anthony.olszanski@fccc.edu; 2Department of Pathology, Fox Chase Cancer Center, Philadelphia, PA 19111, USA; muthanna.jabbar@tuhs.temple.edu (J.M.); yuelynn.wang@fccc.edu (Y.L.W.)

**Keywords:** recurrent melanoma, BRAF mutation, NRAS mutation, chronic lymphocytic leukemia

## Abstract

Melanoma is the fifth most common cancer in the United States. The advent of immunotherapy and molecular targeted therapy has improved progression-free and overall survival in many patients with advanced disease. However, the selection of therapeutic choices requires a nuanced approach, especially when considering molecularly targeted agents. This case report highlights a diagnostic and therapeutic challenge in managing a patient with a history of chronic lymphocytic leukemia (CLL) and recurrent melanoma. Molecular testing suggested discordant BRAF V600E testing and a simultaneous NRAS G12D mutation. After a careful literature review, repetition of his molecular testing, and analysis of the timelines and results of all his molecular testing, we concluded that the BRAF V600E mutation result was falsely positive. The patient was treated with two cycles of ipilimumab (1 mg/kg) and nivolumab (3 mg/kg) as per the NADINA trial and had a complete radiographic response. He then underwent resection demonstrating a pathologic partial response ranging from 20% to 95% tumor necrosis, dependent on the satellite examined. This case report underscores the importance of precise molecular diagnostics in guiding melanoma treatment and demonstrates the complexities of managing a patient with a coexisting malignancy.

## 1. Introduction

Over the last 10 to 15 years, with the introduction of immunotherapy and molecularly targeted therapy, the melanoma treatment landscape and patient prognosis have drastically transformed and improved. Nearly 66 percent of cases with melanoma harbor a *BRAF* mutation at codon 600 with most of these patients having a V600E mutation while the remaining cases are *BRAF* wild type (WT) or harbor a *BRAF* non-V600E mutation [1]. Of the cases with *BRAF* WT melanoma, nearly 15 to 30 percent have an *NRAS* mutation [2,3]. *NRAS* was the first oncogene identified in melanoma in 1983. *NRAS* is responsible for the alteration of a GTPase, resulting in constitutional activation of the *NRAS* oncogene that subsequently activates the MAPK pathway, ultimately leading to uncontrolled cell growth and proliferation [4].

Patients harboring *NRAS* mutations are typically aged 55 or older, have a history of significant ultra-violet (UV) light exposure, and have thicker primary tumors with a higher mitotic rate [5]. Devitt et al. and Ellerhorst et al. also found that *NRAS*-mutated melanoma was associated with inferior clinical outcomes and lower survival rates, though this has been debated in the literature [5,6,7,8]. *NRAS* mutations typically develop during the initial formation of melanoma and are preserved throughout its pathogenesis [8]. *NRAS* and *BRAF* mutations rarely co-exist. Edlund-Rose et al. suggest a 0.7 percent rate of concomitant *NRAS* and *BRAF* mutations [8,9], presumably because mutations in both genes render the same MAPK pathway constitutively active.

Furthermore, patients with chronic lymphocytic leukemia (CLL) are at a fourfold higher risk of developing melanoma; as such, following diagnosis of CLL, patients are recommended to have annual skin checks with dermatology [9]. Most patients with CLL and melanoma are diagnosed with pathologic Stage III disease or lower, and a literature review has demonstrated that melanoma in patients with CLL has variable incidences of *BRAF* and *NRAS* mutations [10]. The mutational status of CLL patients with melanoma was further investigated given potential prognostic and therapeutic indications. Jebaraj et al. identified *BRAF* mutations present in about 2.8 percent of patients with CLL (4 out of 138 patients), whereas Jones et al. found that up to 14 percent (19/221 *BRAF*; 6/221 *NRAS*) of patients with CLL had either a *BRAF* or an *NRAS* mutation prior to the initiation of therapy or during relapse [11,12]. Despite the presence of *BRAF* and *NRAS* mutations in patients with CLL, targeted BRAF-MEK inhibition therapy has not been shown to be effective but may potentially be helpful with prognostication [12].

We describe the case of a patient with a history of CLL on surveillance who was subsequently found to have recurrent Stage IIIC melanoma who appeared to test positive for both an NRAS G12D and a BRAF V600E mutation. Given the discordant results and confusion over his mutational status, the patient presented for a second opinion.

## 2. Case Presentation

A 70-year-old Caucasian male with a history of chronic lymphocytic leukemia (CLL), formerly treated with two courses of fludarabine/rituximab (FR) and ibrutinib (April 2017–October 2018), initially presented to his local dermatologist in July 2022 for a small, slightly hyperpigmented mass on his right ventral forearm. He reported that the lesion was growing for a few months before he sought medical attention. It was non-painful, non-pruritic, and not draining. He subsequently underwent a shave biopsy, which demonstrated cutaneous melanoma, nodular subtype. In September of 2022, he underwent a wide local excision (WLE) and sentinel axillary lymph node biopsy (SLNBx), which demonstrated a 2.2 mm (Breslow depth) melanoma lesion with 7 mitoses per mm^2^ and was without ulceration or lymphovascular invasion. Pathology indicated negative margins. One out of the four sentinel lymph nodes was positive for metastatic melanoma with a metastatic deposit of 6 mm. The final pathologic staging was consistent with Stage IIIB (pT3a pN1a M0). The patient presented to a local medical oncology clinic for discussion of adjuvant therapy for his Stage IIIC cutaneous melanoma. Next-generation sequencing (NGS) conducted by a commercial laboratory (CL #1) at that time was negative for a BRAF mutation but was notable for an NRAS G12D mutation. He received adjuvant nivolumab from October 2022 through to August 2023.

While on adjuvant nivolumab, in July 2023, the patient noticed a new and enlarging nodule on his right ventral proximal forearm, adjacent to the scar of his previously resected melanoma. He underwent a biopsy of this area on 21 September 2023. Pathology was positive for malignant melanoma (Breslow depth 2.2 mm) and was consistent with an in-scar recurrence. He underwent a restaging PET CT, which was negative for metastatic disease but did reveal a separate and distinct proximal right forearm lesion, consistent with a satellite lesion. The recurrence biopsy was sent out to a different commercial laboratory (CL #2) and underwent targeted mutational testing that demonstrated the presence of a BRAF V600E mutation with no mention of the NRAS status. Due to this finding, the oncologist recommended starting targeted therapy with a BRAF/MEK inhibitor combination.

Given the recurrence of melanoma while on adjuvant nivolumab and the discrepancy in BRAF testing, the patient presented at our institution seeking a second opinion. The satellite lesion was tested again in our laboratory by single-gene Sanger sequencing (SGSS) of BRAF and NGS testing utilizing a panel of 275 genes, including a control blood sample. The BRAF mutation was not detected by either method, and NGS again showed the presence of an NRAS G12D mutation (Table 1).

A discussion was held at our institution by two separate tumor boards: the molecular tumor board and the melanoma tumor board. The consensus agreed that the BRAF V600E result (obtained on September 2023) was highly likely to be a false positive because of the following: (1) the same lesion tested later in March 2024 by two different methods did not present the mutation; (2) NGS in July 2022 and March 2024 by two different labs was consistent (i.e., both showed the absence of BRAF V600E and the presence of NRAS G12D); and (3) the presence of an *NRAS* mutation, which is mutually exclusive in 99.4 percent of melanoma cases. NGS testing did not find a lymphocyte mutation in either *NRAS* or *BRAF*, and the tumor boards, therefore, did not feel that the concomitant diagnosis of CLL confounded the testing results.

Medical oncology recommended initiating ipilimumab and nivolumab (1 mg/kg and 3 mg/kg, respectively) as a neo-adjuvant approach per the NADINA trial [13]. The patient did not receive BRAF-MEK inhibitor therapy. The patient had a remarkable clinical response to dual immunotherapy. Neither of the forearm lesions were palpable following cycle 1 of ipilimumab/nivolumab. He had no significant side effects from immunotherapy aside from Grade 1 fatigue. The patient subsequently underwent surgical resection, and the pathologic response was assessed according to Tetzleff et al., who examined guidelines put forth during the International Neoadjuvant Melanoma Consortium meetings in 2016–2017 [14]. The 4 cm epitrochlear mass showed a treatment effect comprised of approximately 20% tumor necrosis and histiocytic inflammation (pathologic non-response; pNR). A right upper arm nodule of 0.8 cm was consistent with a treatment effect, revealing only pigmented melanophages (pathologic complete response; pCR). Finally, a separate upper arm subcutaneous nodule of 1.5 cm demonstrated focal residual melanoma of less than 5% (major pathologic response; MPR). The totality of the pathologic data suggests a pathologic partial response (pPR) [15].

## 3. Discussion

Found in nearly 30 percent of cases with melanoma, *NRAS* oncogenic mutations are the second most common mutation found in melanoma [2,3]. Pathologic *NRAS* driver mutations result in the constitutional activation of the MAPK pathway. Unfortunately, therapies targeting *BRAF* mutations in melanoma or *KRAS* mutations have shown limited efficacy in patients with *NRAS*-mutated tumors. Conversely, the identification of true-positive *BRAF* mutations is critically important, and these can be effectively treated with BRAF-MEKi therapy. In our patient with recurrent melanoma, we questioned whether a BRAF V600E mutation had developed and whether targeting this mutation would be considered in his next line of treatment. We also questioned the role of his concurrent diagnosis of CLL and its influence on his molecular testing results, as well as his melanoma disease course.

There are numerous available methods to identify mutations that malignancies may harbor. Single-gene Sanger sequencing (SGSS) has been traditionally utilized, given its ability to detect acquired mutations in malignant tissue. Though the result turnaround time is faster with SSGS, it is only able to identify point mutations in the gene of interest. On the other hand, while slower to yield results, next-generation sequencing (NGS) can analyze multiple genes in parallel and identify many different variants simultaneously. In our patient, comprehensive NGS testing at an outside commercial laboratory was utilized during his initial diagnosis. An *NRAS* mutation was found, but the sample did not carry a *BRAF* mutation (Table 1). When his disease recurred, the fresh biopsy tissue was sent out for molecular testing to a different commercial laboratory. Surprisingly, this test was positive for a BRAF V600E mutation. There was no mention of an NRAS G12D mutation, though the complete report was unavailable. We were unable to identify the exact methodologic details of testing obtained at the second commercial laboratory. Given that only one gene mutation was reported, we assume that SGSS was utilized at this laboratory. Secondary to the conflicting reports, we repeated his genetic testing at our institution’s CLIA-certified molecular laboratory. DNA NGS testing at our institution revealed a *BRAF* wild-type, *NRAS* G12D-mutation-positive melanoma consistent with his original molecular report, prior to the melanoma (Table 1).

Laboratories vary in their processes for performing NGS, as well as in the sensitivity and specificity rates they achieve for detecting mutations. At our institution, DNA NGS testing is performed on the tumor tissue while incorporating the patient’s blood (i.e., lymphocytes) as a germline-mutation control to reduce false-positive rates for somatic variants. Furthermore, we require a baseline amount of tissue that contains at least 10 percent of tumor cells. Within these sample requirements, our NGS panel has a mutation-detection limit of 5 percent. Other laboratories may have different parameters for identifying mutations and may require an increased or decreased amount of viable tumor cells. More specifically, some laboratories have a mutation-detection threshold of 10 percent and require a higher percent of tumor tissue in the provided sample. The commercial outside laboratory (CL #2) that our patient’s recurrent tissue biopsy was sent to in September 2023 suggests that their NGS sequencing technique requires a minimum volume of 10 mm^2^ or 10 percent tumor content [16]. We are also uncertain whether NGS testing was completed at this outside commercial laboratory number 2 versus SGSS.

Aside from the differences in testing methods and sensitivity levels, a literature review demonstrates that *NRAS* and *BRAF* mutations are mutually exclusive in melanoma. Edlundh et al. found that the rate of co-existence of both mutations in melanoma samples is estimated to be 0.7 percent (2 out of 292 patients) [8]. The hypothesis behind the lack of co-existence involves the pathogenesis of melanoma where either an *NRAS* or *BRAF* mutation occurs during the initial development of melanoma and remains stable throughout the disease course. Furthermore, *NRAS* mutations commonly result in the activation of CRAF, whereas *BRAF* mutations result in activation of BRAF kinase [17]. Both result in the constitutional activation of the same MAPK pathway.

In addition to our concern about the mutational status of the patient’s recurrent melanoma, he also has a concurrent diagnosis of chronic lymphocytic leukemia (CLL). This raises two questions: (1) Could the discordant *BRAF/NRAS* mutation status be due to a lymphocyte mutation at diagnosis or acquired following treatment for CLL? and (2) Does the concurrent diagnosis of CLL affect the efficacy of immunotherapy for melanoma?

Because our molecular lab includes a normal control using blood lymphocytes, we can rule out the possibility that the patient’s CLL harbored either a *BRAF* or an *NRAS* mutation, and, thus, this did not confound the results. Regarding efficacy, the data suggest that patients with CLL may be burdened by inferior efficacy and decreased tolerability. Cass et al. reported a reasonable melanoma overall response rate (ORR) of 41% in patients but noted that patients previously treated with T-lymphocyte-depleting chemotherapy portended worse outcomes compared with untreated patients (21 vs. 51% ORR, respectively) [18]. Similarly, progression-free survival (PFS) and overall survival (OS) were also compromised by prior treatment [18]. Furthermore, a retrospective case–control study conducted by Jobson et al. reviewed 56 cases of melanoma patients with CLL that were matched to 56 cases of melanoma patients without CLL [19]. Jobson et al. found that patients with CLL and melanoma had worse immunotherapy tolerance (43% vs. 7%) and worse melanoma-specific mortality (1.5 times higher than a patient without CLL) [19]. Despite these data, the use of immunotherapy in a patient with *BRAF* wild-type melanoma and, more specifically, the use of combined CTLA-4 and PD-1 checkpoint-inhibition therapy with recurrent resectable melanoma is a rational therapeutic choice. The patient described herein had a rapid and apparent complete clinical response noted after one dose of ipilimumab and nivolumab and, upon surgical resection, had clear evidence of a partial pathologic response.

In conclusion, special attention is needed when examining the genetic testing of patients with melanoma, as treatment decisions may be guided by mutational testing in the recurrent or metastatic setting. Discordant results should always raise suspicion and should be confirmed with a secondary technology. NGS is more sensitive than SGSS, and a normal control can further clarify the results. This report demonstrates a false-positive *BRAF* mutation in a patient with CLL with recurrent melanoma who was offered targeted therapy based on that result. The use of targeted agents in patients who do not harbor a mutation can be harmful [20]. More specifically, the use of targeted agents in patients without a *BRAF* mutation has been shown to have a very poor response rate to treatment of their disease and, extrapolating from lung cancer studies, a decrease in progression-free and overall survival [20,21,22]. Gibney et. al. also described the effect of utilization of targeted therapy causing uncontrolled MAPK pathway activation through heterodimerization between nonmutant BRAF and RAF isoforms leading to downstream uncontrolled proliferation [23]. It is also important to recognize that concomitant *BRAF* and *NRAS* mutations are exceedingly rare in both melanoma and patients with CLL; therefore, it must be confirmed. Using a control sample, we also revealed that the aberrant BRAF-positive result was not due to a rare *BRAF* mutation in CLL. Our patient clearly benefitted from a neo-adjuvant immunotherapy approach despite prognostic variables that can suggest worse outcomes, including primary PD-1 resistance, an *NRAS* G12D mutation, and an immunocompromised state from his concurrent diagnosis of previously treated CLL. 

## Figures and Tables

**Table 1 ijms-26-01029-t001:** Testing time course and results.

Date	Lesion	Test	Laboratory	BRAF ^2^	NRAS ^3^
July 2022	Right ventral forearm	NGS ^1^	Commercial laboratory #1 (CL #1)	Negative	Positive
September 2023	Right ventral forearm proximal to scar	Sanger sequencing?	Commercial laboratory #2 (CL #2)	Positive	Not available
March 2024	Right ventral forearm proximal to scar	Sanger sequencing	FCCC molecular laboratory ^4^	Negative	Not applicable
		NGS		Negative	Positive

^1^ NGS = next-generation sequencing, ^2^ BRAF= BRAFV600E, ^3^ NRAS = NRAS G12D, ^4^ FCCC = Fox Chase Cancer Center.

## Data Availability

Data are contained within the article.

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
