# Peer review of "Recurrent Melanoma in a Patient with Chronic Lymphocytic Leukemia (CLL) Presenting with an Apparent Co-Existing NRAS and BRAF Mutation: A Diagnostic and Treatment Conundrum"

_ijms, 2025, doi:10.3390/ijms26031029_

Round 1

Reviewer 1 Report

Comments and Suggestions for Authors

Berardi et al.'s work highlights the critical importance of well-executed molecular studies in oncology and their potential application in therapy, as treatment decisions often rely on these findings. The manuscript submitted for review is detailed and thoroughly described. However, the authors should consider expanding the introduction and discussion in several areas:

1.    The other authors suggest a higher incidence of these mutations, potentially up to 66%. Please include this information and cite the following source: doi: 10.1111/ced.12015.
2.    Page 3, Lines 126–127: When discussing the limited efficacy of BRAF and KRAS targeted therapies, the authors should note that resistance to these drugs often develops after several months of treatment due to alternative reactivation of the MEK/ERK pathways. They should also cite the following articles: doi: 10.1038/nature09626; doi: 10.1016/j.bbcan.2022.188754; doi: 10.1038/nature09627.
3.    Page 5, Lines 204–205: In the section where they mention that using targeted therapy in the absence of visible mutations can be harmful, the authors should elaborate on this point by describing the associated side effects in greater detail. What specific issues arise from the use of such drugs? They should also include citations to support this discussion.

Reviewer 2 Report

Comments and Suggestions for Authors

The case is presented well

The melanoma occurred post the diagnosis of CLL and whilst there is a discussion on the impact of the CLL on treatment outcomes, there is no discussion on if the CLL and its treatment could have had any impact on the first melanoma occurence, if there is any

Reviewer 3 Report

Comments and Suggestions for Authors

Thank you for the opportunity to review your work. Unfortunately, the purpose of the work does not become understandable: you treated a patient with a discordance in the mutations detected probably due (and remarked by you in the text ) of a false positive due to the test performed in the second laboratory, moreover using an approach already standard in the literature (as reported in the literature). I would also have some questions about how a shave biopsy was performed on a suspected melanovascular lesion and the use of biopsy of multiple lymph nodes.

Reviewer 4 Report

Comments and Suggestions for Authors

Dear authors,  

the case report is interesting, focusing on the difficult management of a patient presenting with multiple malignancies, as chronic lymphocytic leukaemia (CLL) associated with recurrent melanoma.  

With the improvement of therapies, lot of patients can display multiple malignancies and sometimes the right treatment is not straightforward, especially when molecular findings are important for the treatment and can be suboptimal.  

The paper is original, as it would add to the literature a complex case with therapeutical options and lights on molecular aspect of melanoma. 

The methodology used is appropriate and clear 

The quality of data described is good, and the case is well described.  

Conclusions are consistent with the evidence and argument presented 

References are appropriate and sufficient. 

No ethical problems were found  

No inappropriate or potentially libelous language is present. 

Author Response

Thank you very much for reviewing our manuscript. We appreciate the kind words.

Round 2

Reviewer 1 Report

Comments and Suggestions for Authors

The authors responded to the criticism in a proper way. I have no further comments.

Reviewer 3 Report

Comments and Suggestions for Authors

Author have answered my doubts.